# Bioinformatics Tools for NGS-Based Identification of Single Nucleotide Variants and Large-Scale Rearrangements in Mitochondrial DNA

**DOI:** 10.3390/biotech14010009

**Published:** 2025-02-12

**Authors:** Marco Barresi, Giulia Dal Santo, Rossella Izzo, Andrea Zauli, Eleonora Lamantea, Leonardo Caporali, Daniele Ghezzi, Andrea Legati

**Affiliations:** 1Unit of Medical Genetics and Neurogenetics, Fondazione IRCCS Istituto Neurologico Carlo Besta, 20126 Milan, Italy; marco.barresi@istituto-besta.it (M.B.); giulia.dalsanto@studenti.unimi.it (G.D.S.); rossella.izzo@unimi.it (R.I.); eleonora.lamantea@istituto-besta.it (E.L.); andrea.legati@istituto-besta.it (A.L.); 2Department of Pathophysiology and Transplantation (DEPT), University of Milan, 20122 Milan, Italy; 3Laboratory of Neurogenetics, IRCCS Institute of Neurological Sciences, 40139 Bologna, Italy

**Keywords:** bioinformatics, mtDNA, mitochondrial DNA, NGS

## Abstract

The unique features of mitochondrial DNA (mtDNA), including its circular and multicopy nature, the possible coexistence of wild-type and mutant molecules (i.e., heteroplasmy) and the presence of nuclear mitochondrial DNA segments (NUMTs), make the diagnosis of mtDNA diseases particularly challenging. The extensive deployment of next-generation sequencing (NGS) technologies has significantly advanced the diagnosis of mtDNA-related diseases. However, the vast amounts and diverse types of sequencing data complicate the interpretation of these variants. From sequence alignment to variant calling, NGS-based mtDNA sequencing requires specialized bioinformatics tools, adapted for the mitochondrial genome. This study presents the use of new bioinformatics approaches, optimized for short- and long-read sequencing data, to enhance the accuracy of mtDNA analysis in diagnostics. Two recent and emerging free bioinformatics tools, Mitopore and MitoSAlt, were evaluated on patients previously diagnosed with single nucleotide variants or large-scale deletions. Analyses were performed in Linux-based environments and web servers implemented in Python, Perl, Java, and R. The results indicated that each tool demonstrated high sensitivity and specific accuracy in identifying and quantifying various types of pathogenic variants. The study suggests that the integrated and parallel use of these tools offers a significant advantage over traditional methods in interpreting mtDNA genetic variants, reducing the computational demands, and provides an accurate diagnostic solution.

## 1. Introduction

Primary mitochondrial disorders (PMDs) include a group of defects of mitochondrial biogenesis or function, caused by inherited or sporadic pathogenic variants in mitochondrial DNA (mtDNA) or in nuclear-encoded genes necessary for mitochondrial function, such as the enzymes of the respiratory chain (RC).

Human mtDNA exhibits unique genetic and biological features that make diagnosing PMDs particularly challenging [1]. In its native state, mtDNA is a double-stranded circular DNA molecule, lacking histones and present in multiple copies per cell. This leads to an increased mutation rate compared to nDNA. Most mtDNA genetic defects associated with PMDs are attributable to short variants, such as single nucleotide variants (SNVs), INDELs, and large-scale rearrangements (LSRs), which are most often associated with single large-scale deletions [2]. mtDNA variants can be either in a homoplasmic state, where all copies of mtDNA in a cell carry the same mutation, or in a heteroplasmic state, where only some copies have mutations, allowing both normal and mutated mtDNA to coexist in a single cell [3,4]. The proportion of pathogenic mtDNA variants can differ between individuals within the same family, and also among different organs and tissues within a single person. This variation helps explain the wide range of clinical symptoms seen in individuals with mtDNA-related disorders [5,6]. Furthermore, the proportion of specific heteroplasmic variants can vary over time. For instance, harmful SNVs or deletions might be diminished in tissues with rapid turnover, such as leukocytes, while some mutations may build up in postmitotic tissues like muscle or brain. Another challenge arises from the presence of NUMTs, non-functional nuclear fragments that derive from mtDNA [7]. NUMTs can cause variants to be misidentified as mtDNA due to their high sequence similarity. Additionally, these fragments interfere with the calculation of variant allele frequency (VAF), which is essential for assessing heteroplasmy [8]. Finally, testing an appropriate tissue for accurate genetic diagnosis is essential, as some pathogenic variants progressively decline in blood with increasing age (e.g., *MT-TL1* m.3243A > G (NC_012920.1)) [9] and others can be restricted to skeletal muscle (e.g., *MT-TP* m.15975T > C, m.16002T > C, m.15998A > T (NC_012920.1)) [10] and large-scale mtDNA deletions [11,12]).

Traditional diagnostic methods for PMDs involve detailed clinical assessment, biochemical and histochemical analysis of RC enzymes, Sanger sequencing of mtDNA, and Southern blotting (SB) or long-range PCR. However, these approaches are less effective at detecting low-level heteroplasmy. These traditional methods are increasingly being replaced by more systematic, unbiased genetics-first approaches based on next-generation sequencing (NGS) technologies. The adoption of NGS-based whole mtDNA genome analysis, whole exome sequencing (WES) including mtDNA probes, or complementary multi-OMICS approaches (RNA-seq sequencing, proteomics and metabolomics) in clinical genetics has significantly improved the diagnosis of mtDNA-related diseases [13,14,15].

The performance of NGS is continually being refined through advancements in bioinformatics; improvements in computational methods have made sequence alignment, annotation, variant calling, and prioritization more accurate and reliable [16]. However, the vast amount and diverse types of sequencing data complicate the interpretation of variants. Performing mtDNA sequencing with NGS requires the use of dedicated bioinformatics tools. These tools must be adapted from those created for nuclear DNA to accurately detect and quantify mtDNA variants. Such customization is essential to address the specific features of the mitochondrial genome throughout the entire process, from sequence alignment to variant calling [17]. There are now several tools and web servers available which are specifically designed for mtDNA SNV detection, identification of INDELs, LSRs, and heteroplasmy calling, but their evaluation has shown highly variable predictive capabilities [18,19,20].

Our study presents innovative bioinformatics methods designed for both short- and long-read sequencing data, with the goal of significantly improving the precision and reliability of mtDNA analysis in diagnostic applications. These advanced approaches enrich the current toolkit by offering additional capabilities for the detection and interpretation of mtDNA variants.

## 2. Materials and Methods

### 2.1. Sample Collection and Preparation for NGS

We analyzed two cohorts, each comprising 10 patients, with different mtDNA defects: single large-scale mitochondrial DNA deletion syndromes (SLSMDSs) [21] (Table 1) and point mutations with different levels of heteroplasmy associated with PMDs such as MELAS and Leigh Syndrome (Table 2). Total DNA was extracted from frozen muscle biopsy samples or blood using standard protocol phenol/chloroform. All defects had already been previously diagnosed using classical sequencing methods (restriction fragment length polymorphism (RFLP) and SB) or more recent approaches based on short-read NGS, including both targeted panels and WES [15] or long-read NGS for mtDNA [22].

### 2.2. Bioinformatic Data Analysis

The distribution pattern and quantity of LSRs (large-scale deletions and duplications) were analyzed accurately using high-throughput computational pipeline MitoSAlt (version 1.1.1) with paired-end Perl script [23]. The MitoSAlt package is open source and available through SourceForge https://sourceforge.net/projects/mitosalt/files/MitoSAl_1.1.1.zip/download (accessed on 15 November 2024). The bioinformatics analyses were conducted using Linux-based work environments built on cluster HPC and requiring pre-installed Perl and R. MitoSAlt relies on several external software packages, which are conveniently installed in the designated MitoSAlt directory by executing the setup.sh script with the command chmod +x setup.sh && bash setup.sh. For input, MitoSAlt needs a configuration file, fastq.gz files from paired-end sequencing, and a specific name for each run, which is used for all output formats. This is achieved by running the following commands:

perl MitoSAlt1.1.1.pl config_human.txt \

sample_name_L001_R1_001.fastq.gz \

sample_name_L001_R2_001.fastq.gz \

sample_name.

The config_human.txt file includes a steps section that allows for the identification of LSRs from the same output file using various parameters, which we adjusted based on our requirements (Table 3). The outputs (cluster and breakpoint files) are then processed by the R script delplot.R (line 203 of the *MitoSAlt1.1.1.pl* script) to filter deletions and duplications based on estimated heteroplasmy, producing the final report data in TSV format and circular plots visualizing the predicted deletions and duplications. The mean runtime for the MitoSAlt analyses is around 5–10 min for data of a single sample from targeted panels and around half an hour for WES data.

The Mitopore [24] pipeline was used to detect SNVs and INDELs, and for haplogroup identification. For all patient samples affected by these types of variants (Table 2), the SNV&INDEL pipeline was run automatically with software developed in Python (version 3.10.6), Java (version opendk 11.0.19), and R (version 4.2.1), freely accessible through the Jango (version 4.1.1) web server (https://mitopore.de, accessed on 15 November 2024). The analysis required some preliminary steps: each pair of file_R1.fastq.gz and file_R2.fastq.gz generated by sequencing was decompressed into fastq format using the command gzip -dk and then merged with the command cat file_R1.fastq file_R2.fastq > sample_name.fastq in Linux. For upload to the web server, sample fastq data can be compressed into a zip archive, whether they are to be analyzed individually or in groups. The reference genomic sequence in rCRS fasta format was also uploaded. After confirming and waiting for the upload to be completed, a success email was sent indicating the start of the run, followed by a second email with a download link for the folder containing the final output. The mean runtime for the Mitopore analyses is around 5 min for data of a single sample from targeted panels. From the web interface, we then selected certain parameters (Table 4) that can be adjusted according to the purpose of the analysis.

Some WES-derived samples, which were very large in size (>1 GB), cannot be loaded directly onto the web server but were instead analyzed using the Mitopore command line tool within a Docker container (https://github.com/thachnguyen/mitopore_workflow accessed on 15 November 2024). The Mitopore Docker image was converted to a Singularity-compatible .sif file to better suit the performance and usage requirements of HPC cluster. The run was initiated with the following commands:

singularity run -B \

/path_to_main_fastq_folder:/mitopore_data \

/path/mitopore_workflow.sif python \

/home/ag-rossi/projects/mitopore_workflow/mitopore_local/mitopore_indel.py /mitopore_data/ illumina.

The files in fastq format were uploaded to a subfolder named fastq within the main directory. The mean runtime for WES analysis of a single sample is around 20–30 min (64 Gb RAM). Even by using the Docker container for some WES data, an error (“out of memory”) hampered the analysis. A bioinformatics downsampling/enrichment step of data containing mtDNA reads was needed and was suitable for web server analysis.

## 3. Results

### 3.1. SLSMDSs Analysis

The evaluation of circular plots across the cohort of patients with SLSMDSs revealed that MitoSAlt precisely identified all final events at single-nucleotide resolution, correctly categorizing them as single large deletions in all the samples tested. Each of these deletions shared the same profile, with lengths ranging from about 1000 to 5000 bp (Figure 1).

Although the mutation appears as a single macrodeletion, MitoSAlt is designed to identify and prioritize the main event, which is the one supported by the highest number of reads covering the breakpoints (alt.reads), along with a few secondary ’single large-scale deletion’ events that differ by only a few bases in the breakpoint positions. Reads spanning the breakpoints may align to nearby positions due to the similarity of the mtDNA sequence, creating additional deletion events as a result of alignment bias. This error increases with higher coverage, leading to more event calls. Progressive downsampling to achieve uniform coverage across samples can enhance the accuracy of the analysis (Appendix A). Nevertheless, the called main event in all tested samples allowed for the definition of the exact breakpoints and the size of the deletion; these features cannot be obtained by Southern blot analysis or long-range PCR, which provide just an approximate indication of the deletion size (see Table 1).

A detailed analysis of the CLUSTER file reports, which organize breakpoints according to a defined distance threshold, along with the final TSV file reports, does not provide direct reporting of heteroplasmy levels; rather, it enables the estimation of heteroplasmy for a specific pair of clustered breakpoints. This estimation is based on the ratio of alt.reads and the sum of alt.reads and the wildtype reads that overlap them (ref.reads).

Additionally, the MT11267M sample, sequenced using ONT long-read NGS, was further analyzed using the dedicated MitoSAlt_SE1.1.1.pl script, designed for single-end data generated by ONT technology. This analysis provided additional confirmation of the algorithm’s effectiveness in detecting macrodeletions in long-read data. The event detection demonstrated high accuracy and precision, allowing MitoSAlt to focus exclusively on identifying primary events while efficiently filtering out the secondary events that often occur in short-read sequencing due to minor variations in breakpoint positions (Appendix A).

### 3.2. SNV Analysis

Mitopore provides a comprehensive solution for detecting clinically relevant mtDNA SNVs. We tested this tool on 10 samples with different mtDNA variants and various heteroplasmy levels. The results in HTML reports are displayed through an interactive CGView SNV plot, a disease variant plot, a table with details on variants, heteroplasmy levels and haplogroup identification. An overview of the Mitopore outputs is reported for sample MT8636M, harboring the m.13513G>A variant in *MT-ND5* (Figure 2). Detailed views of some reports are provided in the Appendix A.

Additionally, plots for sequencing quality control and alignment are available. By zooming in and out on the CGView SNV plot, users can gain a deeper understanding and more effectively classify results, whether they are working with single-sample or multi-sample runs (Figure 2a, Appendix A). Moreover, the level of detected heteroplasmy is displayed for each variant similarly to the output of the Mitoverse tool (https://mitoverse.i-med.ac.at accessed on 15 November 2024) (Figure 2b, Appendix A). Mitopore was able to detect pathogenic variants in all tested samples, and the estimated heteroplasmy was highly overlapping with that observed by IGV visual inspection on NGS data (Table 2; mean difference 0.4%, maximal difference 2%). An interesting feature is the disease variant plot (Figure 2c) which shows all mtDNA variants and highlights likely pathogenic ones based on MITOMAP classification. However, some automatic filtering steps (probably based on the coverage or quality of the reads but not detailed in the Mitopore report) could remove variants of interest from the final plot. Mitopore has demonstrated its ability to accurately identify all variants in a broad range of heteroplasmy up to 98–100% across all tested samples obtained through sequencing with targeted panels (Figure 3). All reported results are based on a variance threshold set at 5% (Table 4), but this can be easily modified down to 1%.

### 3.3. SVs Analysis by Mitopore

To demonstrate the effectiveness and robustness of the Mitopore workflow for identifying structural variants, we analyzed samples from the SLSMDS patient cohort, along with an additional MT11267M sample sequenced using long-read ONT technology, running the beta version of the SNV&INDEL pipeline. For the short-read cohort samples, the tool was unable to efficiently detect individual large deletions, displaying a message in the relevant section stating the absence of large deletion events, despite these being correctly diagnosed with MitoSAlt (version 1.1.1). Although the software currently checks the coverage track and briefly verifies the coverage baseline using a 200 bp threshold, it does not automatically identify these large deletions in short-read data.

Since Mitopore’s pipelines have been optimized for in silico data with reads generated via ONT sequencing, the tool correctly flagged the likely presence of a large deletion in the MT11267M sample (Appendix A).

## 4. Discussion

The accuracy demonstrated by MitoSAlt in identifying single large-scale deletion events in short-read sequencing data, including WES, from patients with mtDNA rearrangements suggests that this tool is highly effective in detecting these events at the single nucleotide level. The precise identification of deletions underscores the reliability of the algorithm, despite the challenges posed by the complexity of mitochondrial genomic sequences.

Although the pipeline does not directly provide estimates of heteroplasmy levels, these have been indirectly inferred based on the number of reads supporting each event and the average depth of mtDNA coverage. This approach has enabled reliable estimates of heteroplasmy, offering valuable insights for the clinical interpretation of mitochondrial variants. The results obtained with MitoSAlt demonstrate that progressive downsampling can significantly enhance accuracy in identifying the exact breakpoints of macrodeletions. Specifically, reducing the initial coverage led to a decrease in the detection of secondary events that differed by just a few bases at the breakpoint positions. This phenomenon suggests that improving data quality through controlled uniform coverage helps mitigate alignment biases, which can be exacerbated by an excessive number of reads in the regions of the breakpoints, thereby considerably enhancing the overall precision of the analysis.

Another strength of MitoSAlt lies in its flexibility in analyzing data from various sequencing technologies. In addition to short-read data, MitoSAlt includes a dedicated script (MitoSAlt_SE1.1.1.pl) for analyzing long-read data obtained from ONT. This script allows the pipeline to process single-end sequencing data with the same level of accuracy and precision, adapting to the specific characteristics of ONT data. This adaptability can enhance the identification of complex structural variants in the mitochondrial genome, providing a more comprehensive and less error-prone analysis compared to short-read data.

Mitopore has proven to be a reliable tool for detecting single nucleotide variants (SNVs) in mtDNA, supporting not only the identification of clinically relevant variants but also offering detailed visualization of results through its comprehensive and interactive reports.

The high accuracy of variant detection at elevated levels of heteroplasmy (98–100%) underscores Mitopore’s robustness for clinical analyses of samples obtained through targeted panels or WES.

However, the inability of Mitopore to automatically detect large deletions in samples sequenced using short-read technology highlights certain current limitations of the SNV&INDEL algorithm in handling real-world data. Despite the presence of preliminary verification of the coverage track, the lack of automatic recognition of deletions indicates the need for future updates to the pipeline. This limitation may stem from the fact that the pipeline was optimized for in silico data, which lack the significant deletion events in short-read data. However, future updates to Mitopore could address this, using real short-read data and potentially applying different alignment scores specifically designed for large INDELs.

Nevertheless, the effectiveness of Mitopore in detecting deletions in one sample sequenced using ONT technology suggests that the algorithm could be refined to support the identification of deletions even in real short-read data. In this context, implementing new alignment parameters specifically designed for large INDELs could significantly enhance the tool’s performance.

Currently, the commonest way to check for the presence of large deletions in NGS data is to manually verify their presence by inspecting the BAM file using IGV or another genome viewer; this is a time-consuming step and clearly requires a suspicion of this kind of mtDNA defect. The possible use of automatized tools like MitoSAlt (or even Mitopore for long-read data) speeds up this type of analysis.

In this study we did not test whole genome sequencing data, but we expect that they can be analyzed in the same way. However, due to their very large size, they likely require some adjustments in the bioinformatics pipeline, such as the use of a Docker container for running the Mitopore command line.

The presence of nuclear sequences or pseudogenes with high homology with the mtDNA (i.e., NUMTs) represents a problem in mtDNA variant calling because NUMT variants can be misidentified as mtDNA due to their high sequence similarity. However, the high copy number of mtDNA over corresponding nuclear loci usually makes possible NUMT miscalling irrelevant; the use of biological samples even more enriched in mtDNA than their nuclear counterparts, such as skeletal muscle biopsies, further reduces this problem. NUMTs are a potential source of contamination, particularly for PCR-based mtDNA amplification [8]. However, the NGS protocol we used did not require any long-range PCRs as initial steps. NUMT fragments have been suggested to interfere with the quantification of heteroplasmy, impacting on VAF calculation. Nevertheless, our previous studies on short-read NGS [15] estimated ≈2% heteroplasmy as a detection limit based on the mean percentage of alternate reads of all the mtDNA positions that can be due to either technical/instrumental errors or biological interfering elements, including NUMTs. Furthermore, the presence of two samples with 100% calling of the alternative base (MT10304M and MT10618L in Table 2) is additional indirect evidence of a negligible contribution of NUMTs on mtDNA variant calling (unless we assume that the same variant is also present in nuclear DNA).

The pipeline used by Mitopore does not contain any specific step to exclude NUMTs. The authors stated that the PCR-based enrichment step with meticulous primer design was enough to prevent contamination by NUMTs [24]. Conversely, the MitoSAlt pipeline relies on an initial alignment of sequencing reads in the nuclear and mitochondrial genome to remove nuclear reads (NUMTs) while retaining mtDNA-mapped and unmapped reads [23].

Additional tools and bioinformatics strategies for the analysis of mtDNA genetic defects have been reported (and well described in a recent review [25]). Some of them were tested on simulated sequencing reads or designed for other purposes (e.g., for somatic/tumor variants). For this study, we selected two free tools for the analysis of mtDNA genetic defects, which can be used through user-friendly websites and do not require advanced bioinformatics skills. We tested them on samples with a previous molecular diagnosis of PMD (either samples with single, large-scale mtDNA deletion or samples with heteroplasmic/homoplasmic mtDNA pathogenic SNV), to evaluate their possible use in genetics laboratories.

## 5. Conclusions

The study suggests that the integrated and parallel use of the MitoSAlt and Mitopore tools offers a significant advantage over traditional methods in interpreting diverse genetic mtDNA variants. These free tools can be used for analyzing targeted mtDNA NGS data or data from either short-read or long-read sequencing of genomic regions containing mtDNA. This approach reduces the computational power required for data processing and provides a precise diagnostic solution easily available for genetics laboratories.

## Figures and Tables

**Figure 1 biotech-14-00009-f001:**
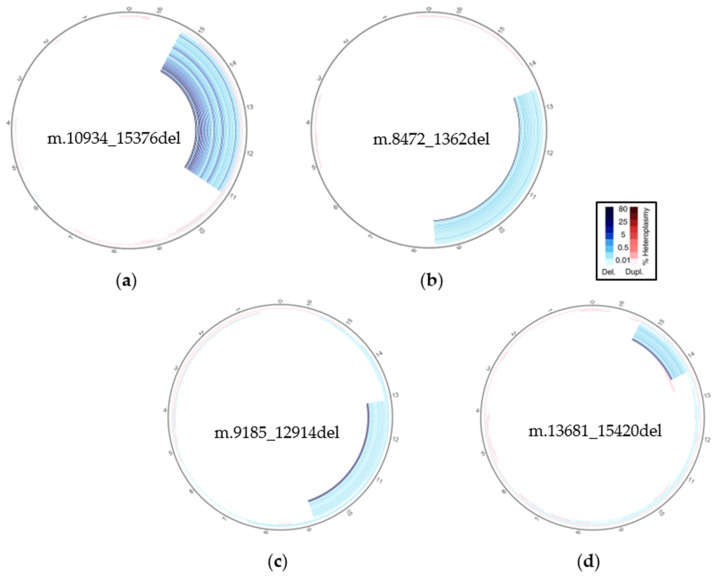
Evaluation of MitoSAlt on patient samples with novel SVs. Circular plots show a single large deletion in MT7093M (**a**), MT9384M (**b**), MT8585M (**c**) and MTT7073M (**d**) selected patients, localized at major arc. The individual arcs are color-coded to represent deletions (blue) or duplications (red), with the intensity of the color reflecting a greater percentage of estimated heteroplasmy, as shown in the legend.

**Figure 2 biotech-14-00009-f002:**
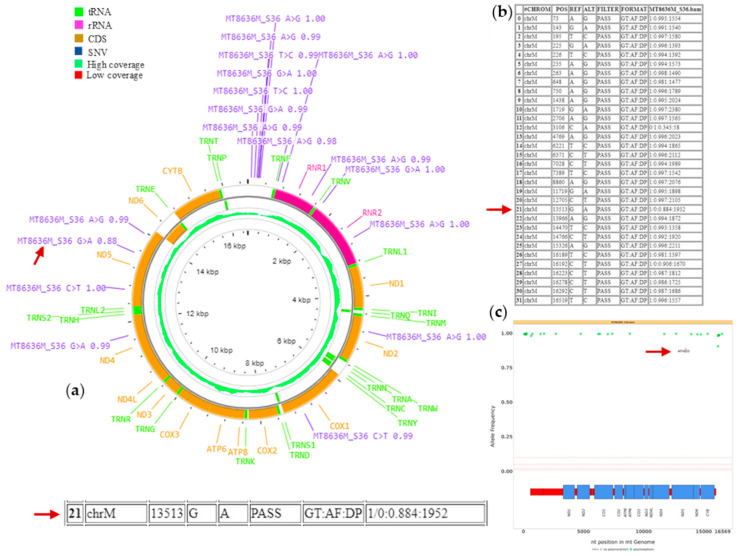
A selection of outputs from the Mitopore.de analysis in an MT8636M patient. (**a**) A CGView SNV plot displays variant calling results in an interactive circular format, with detailed information provided in a table (**b**); the same table is reported at larger size as Appendix A. The analysis distinctly highlights the pathogenic variant marked by the arrow. (**c**) A disease variant plot shows identified variants, annotated using the MITOMAP database. Polymorphic variants are marked in green and potentially pathogenic variants in red, as identified by the tool for the *MT*-*ND5* gene.

**Figure 3 biotech-14-00009-f003:**
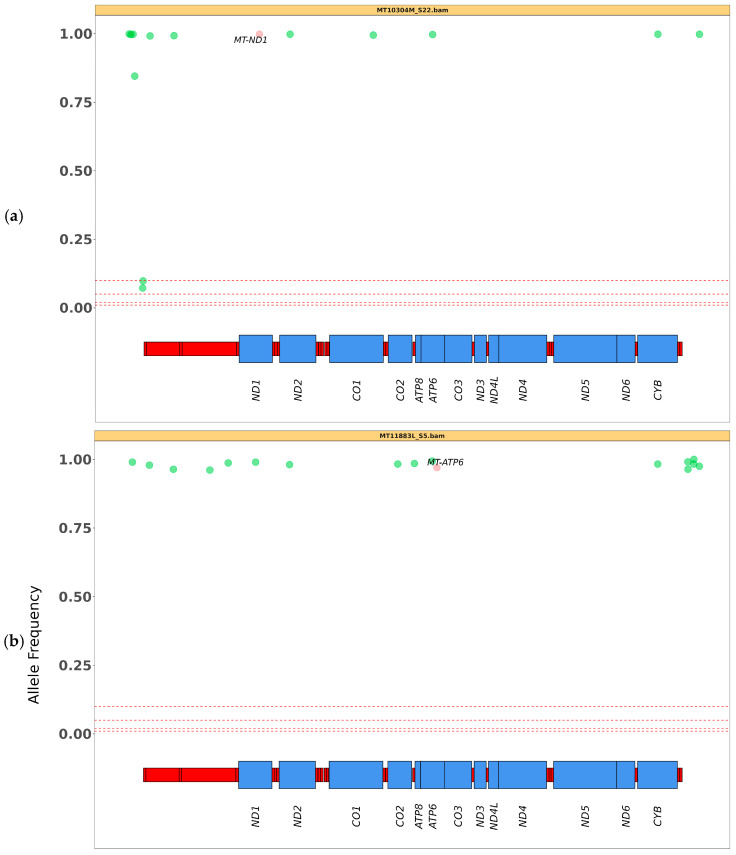
Details of heteroplasmy call reliability for pathogenic variants. The vertical axis shows allele frequency, indicating the detected heteroplasmy threshold at (**a**) 100% (ratio 1.00) for the *MT-ND1* variant, at (**b**) 98% (ratio 0.98) for the *MT-ATP6* variant and at (**c**) 51% (ratio 0.51) for the *MT-TI* variant.

**Table 1 biotech-14-00009-t001:** Cohort of patients diagnosed with SLSMDSs.

Sample	Deletion Size	Heteroplasmy	Deletion Breakpoints (Deletion Size in bp)
MT11227M	~5 kb	~60% ^1^	m.8466_13463del (4980)
MT11267M	~3 kb	~50% ^1^	m.11025_14302del (3270)
MT6297M	~3 kb	~30% ^1^	m.10917_14498del (3555)
MT6885M	~3 kb	~10% ^1^	m.9420_13679del (4259)
MT7073M	~1 kb	~50% ^1^	m.13681_15420del (1732)
MT7093M	~3 kb	~70% ^1^	m.10934_15376del (4425)
MT7321M	~5 kb	~40% ^1^	m.8467_13461del (4980)
MT8585M	~4 kb	~40% ^2^	m.9185_12914del (3721)
MT8903M	~5 kb	~50% ^1^	m.10932_15537del (4594)
MT9384M	~5 kb	~50% ^1^	m.8472_1362del (4980)

^1^—quantification by SB; ^2^—quantification by quantitative PCR; kb—kilobases; bp—base pairs.

**Table 2 biotech-14-00009-t002:** Cohort of patients diagnosed with mtDNA SNV.

Sample ^1^	Pathogenic Variant (Gene)	Heteroplasmy (%)	Heteroplasmy Mitopore (%)
MT11883L	m.8993T>G (*MT-ATP6*)	>95 *–98 **	97
MT8221M	m.9907G>A (*MT-CO3*)	99 **	100
MT10304M	m.3890G>A (*MT-ND1*)	100 **	100
MT9797L	m.3243A>G (*MT-TL1*)	10 *–31 **	31
MT3582M	m.642T>C (*MT-TF*)	90 *–67 **	67
MT7095M	m.4301A>G (*MT-TI*)	50 **	51
MT8636M	m.13513G>A (*MT-ND5*)	80 *–88 **	88
MT8722M	m.3249G>A (*MT-TL1*)	65 **	65
MT10618L	m.11778G>A (*MT-ND4*)	100 **–100 ***	100
MT3289L	m.8993T>C (*MT-ATP6*)	87 **–90 ***	90

^1^—codes with L refer to DNA from white blood cells while codes with M refer to DNA from muscle; *—heteroplasmy quantification by RFLP; **—heteroplasmy quantification by mtDNA NGS; ***—heteroplasmy quantification by WES.

**Table 3 biotech-14-00009-t003:** Configuration parameters utilized for the analysis with MitoSAlt (version 1.1.1).

Mitochondrial Features		Scoring and Filtering Features		Steps	
refchr	MT	score_threshold	80	dna	yes
msize	16,569	evalue_threshold	0.00001	enriched	yes
exclude	5	split_length	15	nu_mt	no
orihs	16,081	paired_distance	1000	rmtmp	yes
orihe	407	deletion_threshold_min	30	o_mt	yes
orils	5730	deletion_threshold_max	30,000	i_del	yes
orile	5763	breakthreshold	−2	cn_mt	no
		cluster_threshold	2		
		Breakspan	15		
		Sizelimit	10,000		
		Hplimit	0.01		
		Flank	15		
		split_distance_threshold	5		

**Table 4 biotech-14-00009-t004:** Selected parameters for Mitopore web server run.

Mitopore Run Parameters	
Type of Analysis	SNV & INDEL
Sequencing data	Illumina (short read)
Genome	Human hg38 (Homo Sapiens)
Haplotree	phylotree-rcrs@17.2
Variance threshold (%)	5
Minimum sequence depth	1000

## Data Availability

Representative raw data are freely available at the repository Zenodo: 10.5281/zenodo.14231615. In addition to data presented in this paper/Appendix A, further data related to this study are available on request from the corresponding author due to privacy/consent reasons.

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
