# Peer review of "Bioinformatics Tools for NGS-Based Identification of Single Nucleotide Variants and Large-Scale Rearrangements in Mitochondrial DNA"

_biotech, 2025, doi:10.3390/biotech14010009_

Round 1

Reviewer 1 Report (Previous Reviewer 2)

Comments and Suggestions for Authors

Summary

This study assesses Mitopore and MitoSAlt, two bioinformatics tools intended to enhance next-generation sequencing investigation of mitochondrial DNA. The study shows how well the tools work to find large-scale rearrangements and single nucleotide variations, providing a more effective option than conventional techniques. Using cohorts of individuals with known mtDNA abnormalities, the authors thoroughly examine their accuracy, specificity, and computing efficiency.

Strengths

The study addresses the challenges of mtDNA variant analysis, an area of increasing clinical relevance. By focusing on both short- and long-read sequencing data, it highlights the adaptability of the tools.

The evaluation is rigorous, including comparisons with traditional diagnostic approaches and detailed descriptions of the computational pipelines.

Both tools are open-source and freely available, enhancing their utility for broader adoption in genetics laboratories.

The authors effectively showcase the capabilities of the tools using clear, well-annotated figures and tables.

Recommendations

Figures such as those displaying circular plots (Figure 1) become redundant and fail to convey additional insights beyond the textual descriptions. Could these be replaced with more dynamic visualizations that integrate data across samples or emphasize key findings?

The paper does not thoroughly compare Mitopore and MitoSAlt with other existing tools. Would benchmarking against widely used alternatives provide better context for their performance?

The authors acknowledge that Mitopore struggles to detect large deletions in short-read data. How do they propose to address this limitation in future updates?

The heteroplasmy results in Figure 3 appear overly simplistic and repetitive. Could additional visualizations to enhance interpretability?

While MitoSAlt includes a pipeline for filtering out nuclear mtDNA fragments, the limitations of Mitopore in this area are glossed over.

While both tools are claimed to be efficient, the authors do not provide detailed benchmarks for runtime or memory usage.

Conclusion

Through the introduction and validation of two bioinformatics tools, this study makes a significant contribution to the field of mtDNA diagnostics. Addressing a few particular shortcomings in the evaluation and visualization techniques used in the article would be beneficial, though. The paper would be much stronger and more suitable for publishing with these changes.

Author Response

>We thank the reviewer for her/his overall positive evaluation and the insightful comments, which allowed us to refine and improve the manuscript.

Recommendation 1

Figures such as those displaying circular plots (Figure 1) become redundant and fail to convey additional insights beyond the textual descriptions. Could these be replaced with more dynamic visualizations that integrate data across samples or emphasize key findings?

>Response: We think it is important to show (and interesting for the audience to see) the main graphical results obtained by MitoSAlt, and figure 1 reports just few examples, highlighting different breakpoints (info added in the revised figure) or heteroplasmy estimation (i.e. number of reads containing the deletion) in different samples.

Recommendation 2

The paper does not thoroughly compare Mitopore and MitoSAlt with other existing tools. Would benchmarking against widely used alternatives provide better context for their performance?

>Response: > We agree that our paper is not a benchmarking study since we did not perform a systematic comparison of all available tools. Our aim was just to suggest a couple of free user-friendly tools for the analysis of mtDNA genetic defects, that can be easily used in genetics laboratories to detect single large-scale mtDNA deletion or heteroplasmic/homoplasmic mtDNA SNV in NGS data.

In the discussion we cited a recent and detailed review reporting different bioinformatics tools for mtDNA analysis.

Recommendation 3

The authors acknowledge that Mitopore struggles to detect large deletions in short-read data. How do they propose to address this limitation in future updates?

>Response: The Mitopore tool was not developed by us but by the group of AG Rossi at the Leibniz Research Institute for Environmental Medicine in Düsseldorf. Actually, it was developed for Nanopore long-read sequencing data. We tested it on short-read data from “positive” samples and it failed to detect the deletion. We have already reported this limitation to the developers of the tool.

Recommendation 4

The heteroplasmy results in Figure 3 appear overly simplistic and repetitive. Could additional visualizations to enhance interpretability?

>Response: This is one of the outputs from the Mitopore tool. As already said for MitoSAlt, we think it is important to show (and interesting for the audience to see) the main graphical results obtained by Mitopore concerning assessment of variant heteroplasmy.

In figure 3 we included representative samples with homoplamic, high-level (98%) or intermediate-level (51%) heteroplasmy variants.

Recommendation 5

While MitoSAlt includes a pipeline for filtering out nuclear mtDNA fragments, the limitations of Mitopore in this area are glossed over.

>Response: We clearly stated in the discussion that “The pipeline used by Mitopore does not contain any specific step to exclude NUMTs” (line 331). However, in the paragraph about NUMTs, we reported the following experimental data supporting the notion that the real impact of NUMTs on mtDNA heteroplasmy calculation is negligible:

- ≈2% heteroplasmy as detection limit, … which can be due to either technical/instrumental errors or biological interfering elements, including NUMTs;

- the presence of two samples with 100% calling of the alternative base is additional indirect evidence of a negligible contribution of the NUMTs on mtDNA variant calling.

Recommendation 6

While both tools are claimed to be efficient, the authors do not provide detailed benchmarks for runtime or memory usage.

Response: According to the reviewer’s suggestion, we added info about the mean runtime for the analyses done using the web-servers Mitopore and MitoSAlt, or the Docker container (for Mitopore analysis of WES samples).

Reviewer 2 Report (Previous Reviewer 1)

Comments and Suggestions for Authors

Major

Please include designs and present adequate quantitative measures and statistical significance, for both bioinformatics tools and traditional methods. This is very important to support claims that the integrated and parallel use of those tools offers a significant advantage over traditional methods in interpreting mtDNA genetic variants, reducing computational demands, and providing an accurate diagnostic solution.

Minor

1) Please check the use of decimal commas with metric units throughout the manuscript. E.g. the Deletion breakpoints (deletion size in bp) in Table 1 and the length of deletions in line 166.

2) Please check and reorganize Table 3 for the use of bold font and vertical lines to separate details.

Author Response

Major comment:

Please include designs and present adequate quantitative measures and statistical significance, for both bioinformatics tools and traditional methods. This is very important to support claims that the integrated and parallel use of those tools offers a significant advantage over traditional methods in interpreting mtDNA genetic variants, reducing computational demands, and providing an accurate diagnostic solution.

>Response: We would like to point up that our aim was to test Mitopore and MitoSAlt on real patient samples with known mtDNA mutations, in order to provide an evaluation of these tools. Their specific technical/computational features and statistics have been reported on the original papers describing them.

We simply tested the two tools on samples (10 samples with single large-scale mtDNA deletion and 10 samples with heteroplasmic mtDNA point mutations) previously diagnosed by different approaches; we tested both short-read and long-read NGS data. In all the samples, the tools were able to detect the previously known genetic defect.

Regarding MitoSalt, we clearly stated that it worked positively on all tested samples (page 5) and outperforms the Southern blot and long-range PCR analyses from a qualitative point of view (page 6). Furthermore, it makes easier and faster the detection of large-scale mtDNA deletions in NGS data compared to the manual inspection currently used (page 9).

Regarding Mitopore, we now clearly stated that it worked positively on all tested samples (page 7) and provide comparative data between the estimated heteroplasmy level obtained by Mitopore and by visual inspection using IGV (table 2). A mean difference of 0.4% was observed, with a maximum of 2%, confirming the accuracy of the tool. An advantage of Mitopore is linked to the multiple outputs it generates in the same analysis (reported in figure 2): each of them can be obtained by other tools (e.g. the mitochondrial haplogroup by Haplogrep, a table of the variants heteroplasmy by Mitoverse…) but the use of Mitopore simplifies and accelerates the diagnostic procedure.

Minor comment 1

Please check the use of decimal commas with metric units throughout the manuscript. E.g. the Deletion breakpoints (deletion size in bp) in Table 1 and the length of deletions in line 166.

>Response: We were not able to find decimal commas in the submitted version. If the reviewer refers to thousands separators, we prefer to avoid commas or points which can be confounding particularly for variant nomenclature (m.3243A>G). Furthermore, in SI (International System of Units) metric, the thousands separator is a half space if possible, while points and commas should not be used for thousands separators.

Anyway, we can adapt the manuscript to mandatory editorial requests.

Minor comment 2

Please check and reorganize Table 3 for the use of bold font and vertical lines to separate details.

>Response: The original table we submitted included vertical lines but it was changed in the current version by the editorial team. Some bold fonts (in the second row) were improperly used and we corrected them.

Round 2

Reviewer 2 Report (Previous Reviewer 1)

Comments and Suggestions for Authors

The revisions made have addressed the key concerns, and the manuscript meets the necessary requirements.

This manuscript is a resubmission of an earlier submission. The following is a list of the peer review reports and author responses from that submission.

Round 1

Reviewer 1 Report

Comments and Suggestions for Authors

Major

Please include designs and present adequate quantitative measures and statistics for both bioinformatics tools. This is very important to support claims that the integrated and parallel use of those tools offers a significant advantage over traditional methods in interpreting mtDNA genetic variants, reducing computational demands, and providing an accurate diagnostic solution.

Minor

1) Please check the use of decimal commas with metric units throughout the manuscript. E.g. the deletion size in bp in Table 1 and the length of deletions in line 163.

2) Please check and reorganize Table 3 for the use of bold font and vertical lines to separate details.

3) In line 146, would it be better to run all data via a Docker/Sigularity image for the consistency of the results? What is the advantage for the separation of data to run through a web server and HPC clusters?

4) Please increase the size of arrow and font used in Figure 2b and 2c.

Reviewer 2 Report

Comments and Suggestions for Authors

Peer Review of the Paper: "Bioinformatics Tools for NGS-based Identification of Single Nucleotide Variants and Large-Scale Rearrangements in Mitochondrial DNA"

This paper introduces two bioinformatics tools, Mitopore and MitoSAlt, designed for analyzing next-generation sequencing (NGS) data focused on mitochondrial DNA (mtDNA) mutations. These tools address the complexities of mtDNA, such as its high mutation rate and the coexistence of wild-type and mutant molecules (heteroplasmy), providing methods to detect single nucleotide variants (SNVs), insertions and deletions (INDELs), and large-scale rearrangements (LSRs). The study presents these tools as valuable for mtDNA diagnostics, comparing their performance against traditional methods and demonstrating their potential in clinical applications.

Strengths:

The study introduces advanced bioinformatics tools tailored for mtDNA analysis, filling a critical gap in precision diagnostics for mitochondrial disorders.

By testing Mitopore and MitoSAlt on real patient samples with known mtDNA mutations, the paper provides a robust evaluation of these tools' performance.

The study details clear steps for setup and analysis, offering transparency in tool application, making it accessible for other researchers and clinicians.

The tools demonstrated high sensitivity and accuracy in detecting mtDNA mutations, with strong implications for improving diagnostic outcomes in clinical genetics.

Both tools are adaptable to short-read and long-read sequencing data, allowing them to be used in a variety of settings and sequencing platforms.

Recommendations:

While Mitopore performs well on long-read sequencing data, it struggles with large deletions in short-read samples. Could the authors explain if they plan to enhance the algorithm to address this gap?

Mitopore provides reliable results only at high heteroplasmy levels (98–100%). Can the tool's sensitivity be adjusted for cases with lower heteroplasmy levels, which are common in clinical diagnostics?

How Mitopore and MitoSAlt compare with other available bioinformatics tools for mtDNA analysis?

How do these tools stand out in terms of accuracy, speed, and ease of use?

The authors mention that nuclear mtDNA (NUMTs) is a potential source of misinterpretation. Could more specific guidance be provided on how these tools handle NUMTs to minimize errors in variant calling?

Conclusion: By presenting Mitopore and MitoSAlt, two tools that show promise in identifying and describing mtDNA variations, this work makes a crucial contribution to mtDNA diagnostics. These tools could have a significant influence on clinical genomics with a few more clarifications and enhancements, especially in managing big deletions in short-read data and improving heteroplasmy sensitivity.